# Randomised controlled trial of a novel online cognitive rehabilitation programme for children with cerebral palsy: a study protocol

Jane Wotherspoon,[1] Koa Whittingham,[1] Roslyn N Boyd,[1] Jeanie Sheffield[2]

¹Queensland Cerebral Palsy and Rehabilitation Research Centre, The University of Queensland, Brisbane, Queensland, Australia
²School of Psychology, The University of Queensland, Brisbane, Queensland, Australia

**Correspondence to**
Jane Wotherspoon;
j.wotherspoon@uq.edu.au

## ABSTRACT

**Introduction** Cerebral palsy (CP) is the most common cause of physical disability in children, with an estimated 600–700 infants born with CP in Australia each year. CP is typically associated with motor impairments, but nearly half of all children with CP also experience cognitive impairment, potentially impacting educational and vocational achievement. This paper reports the protocol for a randomised controlled trial of a computerised cognitive training intervention based on behavioural principles: Strengthening Mental Abilities through Relational Training (SMART). The study aims to investigate SMART's effect on fluid reasoning, executive function and academic achievement in children with CP.

**Methods and analysis** Sixty children with mild to moderate CP (Gross Motor Function Classification Scale I–IV) aged between 8 years and 12 years will be recruited. Participants will be randomly allocated to two groups: SMART cognitive training and waitlist control. Families will access the programme at home over a 4-month period. Assessments will be administered at baseline, 20 weeks and at 40 week follow-up for retention. The primary outcome will be fluid intelligence, while academic achievement, executive function and social and emotional well-being will be secondary outcomes.

**Ethics and dissemination** This study has approval from the Children's Health Queensland Hospital and Health Service Research Ethics Committee (HREC/14/QRCH/377) and The University of Queensland (2017001806). If the computerised cognitive training programme is found to be effective, dissemination of these findings would assist children with CP by providing an accessible, cost-effective intervention that can be completed at home at the individual's own pace.

**Registration details** The study was registered prospectively on 10 November 2017 to present. Recruitment is now under way, and we aim to complete recruitment by June 2019, with data collection finalised by March 2020.

**Trial registration number** ACTRN12617001550392; Pre-results.

## INTRODUCTION

Cerebral palsy (CP), with a prevalence of 1.4 per 1000 live births,[1] is the most common cause of physical disability in children, and an estimated 600–700 infants are born with CP in Australia each year.[2] While CP is typically associated with motor impairments, research is now focusing on accompanying cognitive and executive functioning (EF) deficits and how these impact daily living.[3] It is estimated that approximately 45% of children with CP will have an intellectual impairment impacting significantly on educational achievement.[4–7] A number of studies have found cognitive impairment in CP is associated with long-term difficulties in completing formal education, obtaining competitive employment and living independently.[8–11]

While there is growing awareness of cognitive and EF limitations, interventions for CP are typically associated with improving physical activity, limb function and participation in daily living activities.[12] Few interventions target cognitive function and academic abilities.[12 13] A review looking at new technologies in the treatment of CP

### Strengths and limitations of this study

► This is the first study to trial the effectiveness of a computerised cognitive intervention in maximising fluid intelligence in children with cerebral palsy.
► It is the first randomised controlled trial of a computerised cognitive intervention based on relational frame theory for children with a developmental disability.
► Interventions for cognitive impairment in this population are not readily available, and if effective, this intervention would provide a cost-effective, easily accessible intervention.
► All participants will receive access to the computerised cognitive training intervention prior to the end of the study.
► No active control group is included in this study; therefore, we cannot determine impact of the intervention independent of potential placebo or expectancy effects arising from focused use of a computer program.

and developmental coordination disorder found no studies investigating specific cognitive interventions in these groups.[14] A small number of prior studies have examined whether aspects of cognitive function can be improved, through either web-based multimodal therapy ('Move it to Improve it' (Mitii))[15] or a mindfulness-based yoga programme (MiYoga).[16] A randomised controlled trial of Mitii measured visual perceptual skills of participants and found a significant effect of the web-based therapy programme on visual perceptual skill, although concluded that the effect was not of clinical significance.[15] Mak et al[16] found a mindfulness-based movement intervention demonstrated significantly better sustained attention postintervention than a waitlist control group, but no differences were found for other measures associated with cognition, including working memory (WM) and executive function.

For children with CP who live in remote or isolated communities, access to clinic-based interventions is further limited. This study aims to trial a novel online cognitive rehabilitation programme for children with CP targeted to improve intellectual functioning, EF and educational achievement. The programme—Strengthening Mental Abilities Through Relational Training (SMART)[17]—is founded on relational frame theory, which proposes that the development of language and complex reasoning in humans rests on our ability derive relations between stimuli arbitrarily and without direct experience. An online programme designed to train relational framing ability and potentially improve complex reasoning would be a cost-effective, accessible intervention for children with CP.

## CP and cognitive impairment

CP refers to a group of motor disorders, originating in a non-progressive injury or disturbance to the brain.[18–20] These disturbances occur early in development, impacting the foetal or infant brain.[19] CP is associated with various types of brain lesions,[21] arising from many different causes, congenital and acquired, including intracranial haemorrhage, asphyxia, prematurity, low birth weight or infection.[22]

While motor impairment is a defining feature of CP, more recent definitions acknowledge the frequent comorbidities, such as vision problems, epilepsy and cognitive and communication difficulties.[22 23] Fewer studies have focused on cognitive impairment in CP than on motor impairment, but the research that has been undertaken suggests a significant proportion of individuals with CP have impaired cognitive function.[7] A systematic review of rates of impairment in CP found that 49% of children with CP had an intellectual disability, defined as an IQ below 70, while 28% had a severe intellectual disability, with an IQ below 50.[21] Similarly, a population-based study of more than 1100 individuals with CP in Australia found 45% had been recorded as having an intellectual disability, although level of impairment could not be determined for many in this sample.[7] It is recognised that although

CP is a non-progressive disease, the impact of these additional impairments can exert significant influence over a child's development, impacting academic and vocational outcomes, psychological well-being and quality of life,[21] as well as general health.[7]

## Fluid Intelligence

Among most intelligence researchers, there is broad agreement that intelligence is associated with certain abilities, such as problem solving, understanding abstract ideas or capacity for learning.[24–26] One prominent contemporary model, and the most influential at present in the field of intelligence testing, is the Cattell-Horn-Carroll (CHC) theory of intelligence.[27 28] CHC theory offers a system for classifying cognitive ability that has allowed greater consensus in the literature around what is being measured and referred to by terms such as crystallised and fluid intelligence.[28] Fluid intelligence is a measure of the capacity to solve novel problems and reason abstractly, while crystallised intelligence is a measure of comprehension and acquired knowledge.[29]

While in the past, intelligence was considered a stable attribute, more recently, it has been conceptualised as a quality open to change and development.[30] As a result, attention has turned to potential factors that could play a role in determining intelligence, with the aim of developing interventions.[31] Furthermore, by combining technological innovation with proposed models of cognitive plasticity, the possibility of accessible interventions, delivered via computers, iPads or similar devices, has emerged.[14] For example, Løhaugen et al[32] have proposed a randomised controlled trial assessing the efficacy of computer-based WM training in children with CP.

Various online cognitive training programmes have been developed and many of these have targeted WM, an executive function that involves the ability to temporarily hold and manipulate information. Some researchers have hypothesised that gains in WM could transfer into gains in fluid intelligence.[33 34] Research into this area is ongoing, and results are mixed.[33] While Jaeggi et al[34] generated great interest in cognitive training when they reported gains in fluid intelligence after WM training,[35] numerous studies have failed to replicate such transfer effects,[36] and research continues in this field.

## SMART program

The novel online cognitive training programme to be trialled, SMART, shows promise in pilot studies in helping improve children's cognitive skills (eg, ability to learn, think and reason).[17] SMART is a web-based cognitive training program, currently available in English or Dutch, that directly trains the relational abilities thought to be foundational to complex cognition. SMART is grounded in contextual behavioural science, specifically Relational Frame Theory,[17] which proposes that all human language and complex cognition is underpinned by our ability to relate stimuli arbitrarily. That is, humans can relate stimuli in a manner that does not correspond to the

physical properties of the stimuli. Such relations between stimuli are called relational frames, and a number of such frames exist, such as coordination (eg, same as), comparative (eg, more/less) and temporal frames (eg, before and after).[37] An example of a relational frame of more/less would be with Australian \$1 and \$2 coins, where individuals learn that the \$2 coin is worth more, even though it is physically smaller than the \$1 coin.

Many relational frames, including spatial (eg, under), measurement (eg, bigger) and ordinal (eg, first) are foundational to mathematics.[38] Repeated exposure to spatial, measurement, ordinal and other mathematics-relevant relational frames is considered a critical part of early education, helping students develop their understanding of mathematical concepts.[38]

In addition to learning relational frames, people can also learn behavioural responses through derived relations, not merely through direct experience.[17] Through derived relations, our relational framing abilities greatly enhance our capacity to learn and to interact effectively with others and our environment. As an example, a person might learn that the star Vega is closer to Earth than Canopus, but further than Sirius. They could then derive that Canopus is the furthest star from Earth of the three, even though it has not been directly taught, and Canopus appears brighter than Vega. As relational framing is foundational to complex cognition, the direct training of relational framing itself has the potential to have wide-reaching effects on cognition.

The SMART programme consists of 55 modules that can be worked through at the participant's own pace. Progress to each module requires successful completion of the preceding one. A maximum of five modules can be completed per day. Each module presents a proposition in the form of a relation between non-sense words, and then asks a yes/no question based on the proposition. For example, 'SAJ is the same as MIS. Is MIS the same as SAJ?'. Derived relations are also trained through the addition of more than two non-sense words. For example, 'SAJ is the same as MIS. QUW is the same as SAJ. Is QUW the same as MIS?'. Each module provides multiple examples of the relationship being trained, and if 16 questions are answered correctly, each within a 30 s time frame, the next module is unlocked. Additional relations trained include *opposite*, *more than* and *less than*.

A feasibility pilot study of SMART was conducted over a 9-month period with eight children aged 11–12 years old who had been experiencing educational difficulties at school.[17] Seven of the eight children showed significant increases in their intellectual functioning (as measured by the Wechsler Intelligence Scale for Children – Fourth Edition), an improvement of more than one SD. A 2016 study found similar significant increases in IQ, with a sample size of 15 children aged 11–12 years.[31] While promising, practice effects need to be accounted for when repeated administration of standardised measures of intelligence occurs, as they may influence performance, with average gains of 6–7 points over a 1-month

period found for the Wechsler Intelligence Scale for Children – Fifth Edition (WISC-V) measure of full-scale IQ.[39] Furthermore, assessment of fluid reasoning ability may be more affected by practice effects than verbal or WM tasks,[39] as fluid reasoning tasks are associated with ability to solve novel problems. A randomised controlled design rather than preintervention and postintervention studies could control for practice effects, but the two studies that have used this design have been limited by small sample sizes and high attrition rates.[40 41] To date, no studies have investigated the efficacy of SMART in the CP population.

The underlying cognitive skills trained in the programme are required for vocabulary acquisition, mathematical reasoning, and other academic and learning skills. The programme can be accessed in the client's home at their convenience via iPad, Mac or PC with internet access and can be completed at the child's own pace. As such, it is a potentially cost-effective solution that can be delivered to children with CP who are unable to access ongoing rehabilitation services.

## METHODS

### Specific aims

The aim of this randomised controlled trial is to test the efficacy of a novel web-based cognitive rehabilitation programme for children aged between 8 years and 12 years old with mild to moderate congenital CP.

### Hypotheses

The primary hypothesis to be tested is that in a randomised controlled trial for children aged 8–12 years with CP:

1. Participants in the intervention group will demonstrate improved performance on a standardised test of intellectual ability immediately postintervention when compared with a waitlist control group receiving care as usual (WISC-V).[42 43]

The secondary hypotheses to be tested are that the SMART intervention group will demonstrate improvements in the following outcomes:

1. Academic achievement (Wechsler Individual Achievement Test Third Edition (WIAT-III)).[44]
2. Executive function (Behavior Rating Inventory of Executive Function (BRIEF)).[45]
3. Social and emotional functioning (Strengths and Difficulties Questionnaire (SDQ); Behavior Assessment System for Children – Third Edition (BASC-3); and Social Communication Questionnaire (SCQ)).[46–48]
4. Attention (Conners – Third Edition (Conners-3)).[49]
5. Quality of life (Cerebral Palsy Quality of Life – Child (CP-QOL)).[50]

### Study design

The study is a randomised controlled trial design with waitlist control group to determine the effectiveness of the SMART programme for children with CP aged 8–12 years (see Consolidated Standards of Reporting Trials flow chart in figure 1.) After baseline assessment,

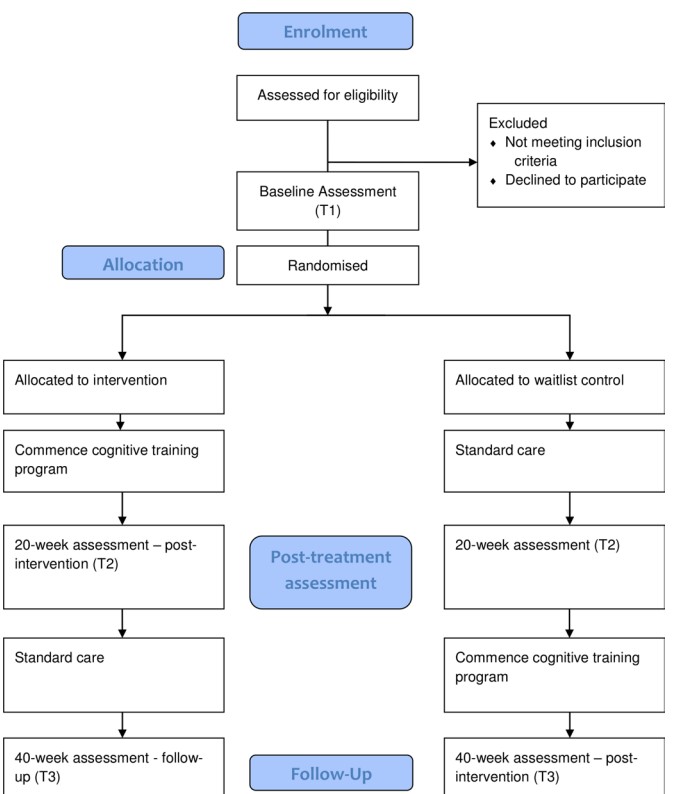

**Figure 1** Flow chart of SMART trial design. SMART, Strengthening Mental Abilities through Relational Training.

participants will be randomly allocated to either the intervention or waitlist control group.

Outcomes will be assessed for all participants at three timepoints, baseline, at 20 weeks postbaseline and at 40 weeks postbaseline. The intervention group will commence the SMART programme immediately and complete it over the following 20 weeks, before undergoing postintervention assessment at 20 weeks postbaseline and follow-up assessment at 40 weeks postbaseline. Participants will be provided with log-in details enabling them to access to the online programme at no cost via the programme website (http://raiseyouriq.com/) for up to 5 months. They will be able to access it at home via either iPad or computer. The waitlist control group will continue care as usual for 20 weeks before returning for a second assessment visit, at which point they will commence the SMART program. Final assessment for the waitlist control group will be postintervention at 40 weeks.

## Participants

We aim to recruit 60 children with mild to moderate CP (Gross Motor Function Classification Scale (GMFCS) – I–IV) aged between 8 years and 12 years old. All children are required to have sufficient cooperation and cognitive understanding to perform the tasks and access the novel online cognitive training program, have access to the internet at home and be able to attend three assessment sessions in Brisbane. Sufficient cooperation and cognitive understanding will be confirmed at baseline assessment,

as participants who are able to undertake the iPad-based assessments will be deemed to meet criteria. Children will be excluded if they have unstable epilepsy (ie, not controlled by medication); an unstable brain injury (eg, degenerative or metabolic condition); and/or active medical condition (eg, chemotherapy, radiotherapy or neurosurgical).

## Recruitment

Participants will be recruited from a consent-based research database at the Queensland Cerebral Palsy and Rehabilitation Research Centre and through the Queensland Paediatric Rehabilitation Service at the Queensland Children's Hospital. Participants will be enrolled in the study by the first author.

## Patient and public involvement

Participants, families and the public were not involved in the design or recruitment of this study. Participants are informed of the study burden from the time of initial contact and are advised of their ability to withdraw from the study at any time. All participants will receive feedback on the results of the assessments administered once they have completed their involvement in the study.

## Randomisation

Baseline assessments and demographic questionnaires will be completed prior to randomisation. Once complete, participants will be randomised to either waitlist control or intervention group. Randomisation will be via stratified random blocks, using a computer-generated block randomisation sequence, stored by staff members unconnected with the study. Allocation to either waitlist control or intervention will be recorded on pieces of paper, and these will be folded, then placed inside opaque envelopes by a staff member not involved in the study. Envelopes will be sealed and only opened on completion of baseline assessment. Participants will be stratified according to IQ ($<70$ or $\geq70$), as measured on baseline assessment.

## Blinding

Given the nature of the intervention, participants will not be blinded as to which group they are assigned to. As assessment will be undertaken by the first author as part of a PhD project, assessors will not be blinded in this project.

## Adverse events

No health or safety risks associated with participation in this study have been identified, and the risk of adverse events is considered low. Any events associated with either intervention or waitlist control groups over the course of the study will be recorded. All participants allocated to the waitlist control group receive access to the intervention after the second assessment. This ensures no adverse impacts through omission of intervention in the event that the intervention is found to be efficacious. If any adverse events are identified, they will be reviewed by the investigators. As the ethical review process and trial

conduct is overseen by two ethics committees, no additional data monitoring is considered necessary.

## Data management

Participants will be allocated randomly generated identification codes, and these will be used to deidentify hardcopy and electronic files. Paper copies relating to assessment will be deidentified and physically stored in a locked filing cabinet at the Queensland Cerebral Palsy and Rehabilitation Research Centre. Electronic data will be stored on REDCap, a secure web platform for creating and managing online databases. The installation of REDCap used for this project is hosted bv the University of Queensland and managed by the Queensland Clinical Trials and Biostatistics Centre.

## Measures

Demographic information will be obtained via a parent survey, gathering information on the participant's background, including gestational age, comorbid diagnoses and GMFCS classification. Further demographic information includes school year, type of school and whether any additional teaching support is accessed, along with parent education and household income.

All children will undergo a comprehensive cognitive, psychoeducational and psychosocial assessment by a psychologist at baseline (ie, before treatment) and reassessment at 20 weeks (after intervention for immediate group) and 40 weeks (retention for immediate group and after intervention for control group). It is noted that many of these assessments have not been validated for children with CP and have been chosen as no valid alternatives are available. However, a review of assessments for children with CP found that motor involvement, communication and visual impairment were key factors in determining suitability of assessments.[51] We have specifically chosen an iPad-based assessment delivery format for our primary outcome (full-scale IQ), that is similar in motor and language demand to the intervention programme itself. If children are able to meet the inclusion criteria for the study, it is anticipated that they will also be able to complete the assessments.

Children will complete the following assessments:

▶ The WISC-V is a standardised measure of overall cognitive/intellectual functioning. The WISC-V produces a Full-Scale IQ Score (FSIQ), along with five primary index scores—Verbal Comprehension Index, Visual-Spatial Index, Fluid Reasoning Index, Working Memory Index and Processing Speed Index, each derived from two subtests. The WISC-V has been found to have good internal consistency ($\alpha$=0.96 for FSIQ and $\alpha$=0.86–0.94 for primary index scores).[43] Inter-rater reliability has also been found to be acceptable ($r$=0.98-.99 for interscorer agreement on a subset of subtests).[42]

▶ The WIAT-III is a standardised measure of educational achievement. It provides measures of achievement across domains of oral language, reading, written

language and mathematics. The internal consistency coefficients are good ($\alpha$=0.83–0.95), with the exception of a single subtests (Alphabet Writing Fluency, $\alpha$=0.69), which is not used with the age range participating in this study.[44]

▶ The SDQ – Child Self-report. This questionnaire measures the child's behaviour and adjustment. The SDQ is a 25-item measure with frequency of behaviour rated on a 3-point Likert scale.[46] The SDQ produces five subscales: emotional symptoms, conduct problems, inattention/hyperactivity, peer problems and prosocial behaviour (range 0–10) and a total difficulties score (range 0–40). Research suggests that for younger children, internal consistency is acceptable for the total difficulties, emotional symptoms, prosocial behaviour and inattention/hyperactivity subscales, but not for peer and conduct problems.[52] However, Muris et al[52] note that the scale can provide useful information about psychopathology in children from 8 years old.

Parents will also be administered questionnaires during the three assessments points. These include:

▶ The BRIEF assesses the child's EF in everyday life. The BRIEF is an 85-item parent-rated questionnaire assessing behavioural manifestations of executive functions.[45] The BRIEF produces a Behavioural Regulation Index (BRI; initiate, WM, plan/organise, organisation of materials,and monitor subscales) and a Metacognition Index (MCI; inhibit, emotional control and shift subscales). The BRIEF has been shown to be a valid measure of EF and has good internal consistency ($\alpha$=0.80-.98) and high test–retest reliability on the BRI ($r$=0.92), MCI ($r$=0.88) and the Global Executive Composite ($r$=0.86).

▶ Conners-3 Parent Report, which assesses childhood behaviours and behavioural disorders including attention deficit/hyperactivity disorder in children and adolescents aged 6–18 years.[49] The Conners-3 consists of 110 statements and takes approximately 20 min to complete. The Conners-3 measures seven key areas: inattention, learning problems, aggression, family relations, hyperactivity/impulsivity, EF and peer relations. Both internal consistency coefficients ($\alpha$=0.83–0.94) and test–retest reliability ($r$=0.52–0.94) are good.

▶ The BASC-3 measures children's adaptive and problem behaviours and emotional difficulties at home and in community settings.[48] It is completed by a parent or guardian and consists of 175 statements, taking approximately 20 min. The BASC-3 produces five composite scales: internalising problems, externalising problems, school problems, adaptive skills and the behavioural symptoms index. Internal consistency coefficients for the Parent Rating Scale ($\alpha$=0.79–0.97 for children aged 6–7 years; $\alpha$=0.79–0.96 for children aged 8–11 years) and test–retest reliability ($r$=0.80–0.92 for Child forms) are good.

- SCQ – Current screens for communication skills and social functioning difficulties associated with autism spectrum disorders. The SCQ – Current evaluates communication skills and social functioning in children, based on the previous 3 months. While research suggests the SCQ – Current has lower internal consistency than the Lifetime version,[53] concerns around its use particularly focus on children below 5 years of age, which is outside the age range of participants in this study, and on using the Current form as an alternative to the Lifetime form for screening for Autism Spectrum Disorder. For the purpose of measuring change over time, however, the Current form is recommended.[47] It contains 40 items and is completed by a parent or primary caregiver in under 10 min. Scores on the SCQ provide information on the following domains: social interaction, communication and stereotyped, repetitive or restricted behaviour.

- SDQ – Parent Report measures prosocial, internalising and externalising behaviours in children. The SDQ is a 25-item parent-report measure of child behaviour and adjustment, with frequency of behaviour rated on a 3-point Likert scale.[46] The SDQ produces five subscales: emotional symptoms, conduct problems, inattention/hyperactivity, peer problems and prosocial behaviour (range 0–10). It produces a total difficulties score (range 0–40) found to have adequate internal reliability ($\alpha=0.73$) and test–retest reliability ($r=0.62$) as well as discriminant and concurrent validity.[46]

- CP-QOL-Child measures child quality of life. CP QOL-Child is a 66-item parent-report measure of child quality of life specifically developed for use in children with CP.[50] It measures quality of life across domains including physical well-being, social well-being, emotional well-being, school, service access and social acceptance. It has good concurrent validity, internal consistency ($\alpha=0.76$–0.89) and test–retest reliability ($r=0.80$–0.90).[50]

The SMART programme provides an internal measure of relational ability, and all children will complete this measure at baseline, 20 weeks and 40 weeks. In addition, information will be recorded on time taken for each participant to complete the programme. Assessments at baseline, 20 and 40 weeks will take approximately 2–3 hours per session. The waitlist control design ensures all children in the study will receive the intervention within 6 months of being randomised to either commence the programme immediately or after 20 weeks. Box 1 summarises the measures for each assessment point.

### Qualitative interview

At the conclusion of the study, semistructured interviews will be conducted with children and caregivers by the first author, a registered psychologist. The aim of the interview is to explore participants' engagement with the online cognitive rehabilitation programme and gain qualitative insights into families' experience with the programme.

---

**Box 1    Summary of assessment measures.**

**Measures (T1, T2 and T3).**
**Child**
Intellectual functioning (Wechsler Intelligence Scale for Children – Fifth Edition).
Academic achievement (Wechsler Individual Achievement Test - Third Edition).
Social-emotional functioning (Strengths and Difficulties Questionnaire – Child self-report).
Relational Ability Index (RAI).
**Parent**
Executive function (Behavior Rating Inventory of Executive Function).
Attention and behaviour (Conners – Third Edition).
Social-emotional functioning (Behavior Assessment System for Children – Third Edition and SDQ – Parent report).
Social communication (Social Communication Questionnaire – Current).
Quality of life (Cerebral Palsy Quality of Life – Child – Parent report).

---

Questions will cover what families liked and disliked about the programme, how easy they found it was to access at home and to remain engaged. If the programme is found to be effective, such qualitative insights will be valuable in the translation phase. The script for the interview can be found in online supplementary appendix A. Interviews will be recorded, transcribed and analysed.

### Intervention

With the SMART online platform, participants answer problems directly training relational framing and receive immediate feedback on their answers. Study participants are provided with an alias login name (to maintain confidentiality) and a password. Children are encouraged to complete the SMART intervention for 30 min per session, for a total of 1.5 hours per week. SMART is incremented and can be completed at the child's own pace. The full dosage is reached when the child completes the entire intervention, which is expected to occur within 20 weeks. Throughout the randomised controlled trial, a psychologist will regularly contact the family and monitor that child's progress, supporting both parent and child in meeting their goals for SMART and maintaining engagement.

A resource guide prepared for this study will be provided to all families. This guide provides technical information on accessing the programme, a description of how to work through each stage of the programme and information on how parents can support their child to work on the programme, including a visual chart to keep track of progress.

### Statistical analysis

Study hypotheses will be analysed by means of appropriate statistical tests, with statistical significance for all tests set at $p<0.05$ with adjustment for multiple comparisons, and all analyses will be intention to treat. We propose to carry out a Benjamini-Hochberg procedure to control for false discoveries.[54]

Mixed analysis of covariance analyses will be conducted with time (baseline and 20 weeks) as the within subjects variable, group (intervention or waitlist) as the between subjects variable and baseline data as the covariate. Secondary analysis will profile cognitive change over time for participants based on their test scores. This will include t-tests and linear regression to explore within-subject differences from postintervention to follow-up and over three timepoints (baseline, postintervention and follow-up) for participants in the intervention group.

### Power analysis and sample size

Power analysis was conducted using the software package G*Power,[55] for an analysis of covariance repeated measures, between factors test. With a sample size of 60 subjects, and assuming an error rate of 5% and p=0.70 for within-subject correlations, this sample size results in 81.37% power to detect a large (Cohen's $f$=0.40) mean difference of 12 IQ points between groups, after allowing for an attrition rate of 10%.

### Ethics and dissemination

Protocol modifications and amendments will be submitted to the ethics committees for approval. Amendments to the protocol will be included in publications reporting on outcomes of this study. All families will be provided with a written informed consent form by the first author at the initial visit (online supplementary appendix B) that they will be required to sign before commencing participation. This trial has been registered with the Australian New Zealand Clinical Trials Registry. Study results will be disseminated through publication in scientific journals and participation in conferences. The authors of this protocol will be authors of any publications describing study outcomes, and professional writers will not be used. Families who participate in the study will receive information on the study results, as well as a feedback report on the outcomes of assessments their child has completed. If the computerised cognitive training programme is found to be effective, dissemination of these findings would assist children with CP by providing an easily accessible, cost-effective intervention that can be completed at home at the individual's own pace.

### Trial progress

This protocol is Version 3, 10 May 2019. The study is currently actively recruiting participants, after initial recruitment commenced in June 2018.

### DISCUSSION

This protocol paper has reported the background and study design for a randomised controlled trial investigating the effectiveness of a computerised cognitive training programme for children with CP. This programme has not previously been studied in this population. The research study will assess children's cognitive skills, executive ability and social and emotional functioning, with

fluid intelligence the primary outcome of interest. Qualitative information will be gathered on families' experience engaging with the programme. Results of the study will be disseminated through peer-reviewed journals and at relevant scientific conferences.

Nearly half of all children with CP are estimated to also have an intellectual impairment, impacting academic achievement and ability to achieve educational and vocational goals in the long-term. If this computerised cognitive training programme is found to be effective, a flexible, easily accessible intervention will be available for this population, where at present there are few options available for addressing difficulties with cognitive skills in CP.

**Contributors** The study was designed and established by all the authors. JW is responsible for the ethics application and reporting. JW is responsible for recruitment and data collection. JW will take a lead role in preparing publications on the clinical outcomes of the study. KW, RNB and JS will contribute to the preparation of publications and are providing supervision throughout the study. JW will take on a lead role for statistical analysis. JW drafted the final version of this manuscript, while all authors critically reviewed and approved the final version. JW will use data from this study to contribute to her PhD thesis.

**Funding** JW is a PhD scholar funded by an Australian Government Research Training Program Stipend and Queensland Cerebral Palsy and Research Rehabilitation Centre Top-up Scholarship.

**Competing interests** None declared.

**Patient consent for publication** Not required.

**Ethics approval** This study has received full ethical approval from the Children's Health Queensland Hospital and Health Service Research Ethics Committee (HREC/14/QRCH/377) and The University of Queensland (2017001806).

**Provenance and peer review** Not commissioned; externally peer reviewed.

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
