## [Reviewer comments · BMJ Open]

ARTICLE DETAILS

TITLE (PROVISIONAL)	A randomised controlled trial of a novel online cognitive rehabilitation program for children with cerebral palsy: A study protocol
AUTHORS	Wotherspoon, Jane; Whittingham, Koa; Boyd, Roslyn; Sheffield, Jeanie

VERSION 1 - REVIEW

REVIEWER	Michelle Jackman John Hunter Children's Hospital, Newcastle, Australia
REVIEW RETURNED	13-Jan-2019

GENERAL COMMENTS	Thank you for the opportunity to review this protocol. This is a fantastic, accessible intervention that could have a great impact for children with CP. Good luck with your study and I look forward to reading the published results in future. Abstract – methods, paragraph should not begin with a number, change to “Sixty”. Intro/background – discuss any existing cognitive intervention in this population, you have mentioned there are few, but have not provided any details about these. Page 4, lines 17-19 – please provide references for the statement “This translates to long-term difficulties associated with failure to complete formal education, obtain competitive employment and live independently”. If there is not currently research that supports this carry over it is important to clearly state that these are a possibility, given that intellectual ability may not be the independent variable that lead to poorer employment outcomes or the ability to live independently. How has language ability been taken into consideration, if the intervention is grounded on theory related to development of language? Need to consider and discuss how physical impairment may impact on ability to accurately assess intellectual ability, particularly for those with more significant physical impairment. Page 6, lines 30-36: “Furthermore, by combining technological innovation with proposed models of cognitive plasticity, the possibility of accessible interventions, delivered via computers, iPads or similar devices, has emerged.” Provide references /examples. Page 7-8 – I wonder if you could provide a little more detail regarding the SMART program – is it individualised to each child and their current intellectual level or does each child undergo
--

exactly the same program? Is it just a time-based intervention or does the child need to show competence in a given area before being able to progress?

How is the SMART program accessed? Is it something that needs to be purchased by the department administering the intervention? Can it be freely accessed? Is it available in multiple languages?

Page 8 – assessment timepoints don't need to be specified under aims, include this in method.

Page 8-9 : primary hypothesis may be clearer if worded: "participants in the intervention group will demonstrate improved intellectual ability on the WISC immediately post intervention compared to a waitlist control group receiving usual care".

Page 9-10: will demographic information/additional baseline information be collected eg. age, MACS, GMFCS, type of schooling they are attending, socioeconomic info etc. This information should be specified.

If the waitlist control group will begin receiving the treatment following 20 weeks of usual care, how will long term (40 week post Rx) outcomes be compared? This may be very important comparison information in regard to long term benefits of the intervention.

Page 10 – how will "sufficient cooperation and cognitive understanding" be determined? Through standardised assessment or ability to complete baseline assessment? This should be specified so that the study could be replicated.

Page 10, line 33. Make reference to assessment timepoints in Figure 1.

Will information on how quickly the child completes that program be collected? I wonder if this might correlate with benefits from the intervention (motivation may correlate with outcomes) and could be interesting information given the current research regarding motivation and outcomes of intervention.

Ensure all information on the CONSORT checklist are specified – randomisation procedure, allocation concealment, specify blinding of assessors.

Outcome measurement – discuss validity and reliability of these measures in the CP population aged 8-12 years (your study population). If they are not validated in this population, that should be made clear and a note made that no valid alternatives are available. Particularly for primary outcome measure.

Page 14, who will conduct the qualitative interviews ie. Will it be the researcher who administered the intervention or completed assessments, or will it be an independent person? Do you know what qualitative design will be used? If so, it would be beneficial to detail this.

Page 15 – specify which statistical tests are planned

CONSORT – Add in when randomisation occurs. I wonder if the flowchart could be slightly modified so that timepoints and the overall timeframe of the study are slightly clearer – am I correct in assuming that there is 5 assessment timepoints given the waitlist control? T1 is the baseline (both groups), T2 is the post intervention assessment (both groups will be assessed at this timepoint?). Specify that this will be the primary endpoint of the figure. It is then unclear exactly when assessments of each group occur – I think there is probably T3 (comparison post Rx Ax), then T4 (intervention 40 weeks follow up) and T5 (control 40 week follow up). It would be great for these timepoints to be clearly reflected in the CONSORT Flowchart.

REVIEWER	Dr. Matt C. Howard University of South Alabama, USA
REVIEW RETURNED	16-Jan-2019

GENERAL COMMENTS	I was asked to review the submission, "A randomized controlled trial of a novel online cognitive rehabilitation program for children with cerebral palsy: A study protocol." The submission is a proposed trail to test the effectiveness of a computer program in developing the cognitive abilities of children with cerebral palsy (CP). While I am generally familiar with the content area, both regarding the assessment of interventions as well as the rehabilitation of special populations, I must acknowledge that I do not have significant experience assessing trials. Nevertheless, I reviewed relevant material provided by the journal, and I feel that I can appropriately assess this submission. Overall, I felt that this submission achieved the typical objectives of a proposed trial. It was clear, concise, and provided an adequate overview of the proposed study. I do have three suggestions that I believe should be implemented before the submission is accepted, however. 1.) The proposal discusses a wait-list control design. On Page 3, the submission indicates that, "No active control group is included in this study; therefore we cannot determine impact of the intervention independent of focused use of a computer program." Can't the authors determine this using the wait-list control group? I am confused regarding the intent of this statement, and therefore it should perhaps be reworded or clarified. 2.) I believe that a section is needed which describes the SMART program in more detail. Is it a text-based program? Is it a picture-based program? Is it virtual reality? Of course, computer programs greatly vary, and findings regarding one program may have little to no relation with a different computer program. As the submission currently reads, I am unsure regarding which literatures should be incorporated with the proposal trail. It would be most ideal if the authors could upload a video of the program in use as supplemental materials. 3.) The authors use a wait-list control group design, but they cite prior research that convincingly argues that the intervention is effective. Also, as it is a computer program, I would assume that the program can be extended to all participants fairly easily. In other words, the use of a wait-list control group does not seem to be a logistical necessity, and it is instead solely for the test of the intervention. Can the authors provide any assurance that the delayed administration to wait-list participants will not negatively affect them? Otherwise, I would see the use of a wait-list control design as causing undue harm to the participants.
--

REVIEWER	Chris Jones Brighton and Sussex Medical School, UK
REVIEW RETURNED	24-Jan-2019

GENERAL COMMENTS	The objectives of the study are clearly described, but the statistical analysis section needs to be rewritten to make it clear that the sample size and planed analysis fit with the objectives. Information about randomisation and data management are also required.
---

	The primary analysis is a comparison of the intervention and control groups "immediately post-intervention" - i.e. at 20 weeks. No comparison can be made at 40 weeks, because at this point both groups will have received the intervention. I'm not sure what the point of the assessments at week 40 is, as comparing the groups at this point doesn't make sense for the objectives of the study. Despite this, the analysis method suggested is a repeated measures ANOVA using data from baseline, week 20 and week 40. Since the situation at week 40 is different, and week 40 is not mentioned as part of the primary objective, it should not be included in the analysis. A more appropriate approach would be to perform linear regression for WISC-V at 20 weeks, with intervention group and WISC-V at baseline as independent variables. Generally this section needs to be re-written by a statistician for clarity. The sentence "Study hypotheses will be analysed by means of appropriate statistical tests, with statistical significance for all tests set at $p < 0.05$ with adjustment for multiple comparisons, and all analyses will be intention to treat." lumps several important statistical concepts together in a way that adequately describes none of them. What multiple comparisons are being adjusted for? How is this adjustment being made, and was this taken into account for the sample size calculation? The sample size calculation should be described in a separate section. It likely needs to be done differently as it is based on what appears to be an inappropriate analysis method. Not enough information is provided to recreate the current calculation - all assumptions made need to be clearly stated, as does the software used. What is the justification for using an effect size of 0.7? Is this based on any previous data? What effect size would be considered medically significant? What attrition rate is expected? The randomisation method used needs to be stated clearly. It's probably simple randomisation, but it could be stratified by GMFCS for example - it needs to be clear. Also, how was the randomisation list generated? - block sizes, software used etc. How does it work in practice? Data management - no information is given with regards to data management - what database software will be used and where will the data be stored?
--	--

VERSION 1 – AUTHOR RESPONSE

Reviewer(s)' Comments to Author:

Reviewer: 1

Reviewer Name: Michelle Jackman

Institution and Country: John Hunter Children's Hospital, Newcastle, Australia

Please state any competing interests or state 'None declared': None declared

Please leave your comments for the authors below

Thank you for the opportunity to review this protocol. This is a fantastic, accessible intervention that could have a great impact for children with CP. Good luck with your study and I look forward to reading the published results in future.

Abstract – methods, paragraph should not begin with a number, change to “Sixty”.

Thank you. Changed to Sixty. (Page 2)

Intro/background – discuss any existing cognitive intervention in this population, you have mentioned there are few, but have not provided any details about these.

One recent review found no studies looking specifically at cognitive interventions in this population and I have added a reference to this. However, I have also included additional details regarding two studies that looked at elements of cognitive function, including attention, working memory, executive function and visual perception, and noted limited effects were found for most of these outcome measures. (Page 4)

A review looking at new technologies in the treatment of CP and developmental coordination disorder found no studies investigating specific cognitive interventions in these groups. 14 A small number of prior studies have examined whether aspects of cognitive function can be improved, through either web-based multimodal therapy (“Move it to Improve it” [Mitii])¹⁵ or a mindfulness-based yoga programme (MiYoga)¹⁶. A randomised controlled trial of Mitii measured visual perceptual skills of participants and found a significant effect of the web-based therapy program on visual-perceptual skill, although concluded that the effect was not of clinical significance. 15 Mak et al. found a mindfulness-based movement intervention demonstrated significant better sustained attention post-intervention than a waitlist control group, but no differences were found for other measures associated with cognition, including working memory and executive function. 16

Page 4, lines 17-19 – please provide references for the statement “This translates to long-term difficulties associated with failure to complete formal education, obtain competitive employment and live independently”. If there is not currently research that supports this carry over it is important to clearly state that these are a possibility, given that intellectual ability may not be the independent variable that lead to poorer employment outcomes or the ability to live independently.

References of studies of participation of adults with CP in employment have been included to support this statement. (Page 4)

A number of studies have found cognitive impairment in CP is associated with long-term difficulties in completing formal education, obtaining competitive employment and living independently.⁸⁻¹¹

How has language ability been taken into consideration, if the intervention is grounded on theory related to development of language?

Language ability has been taken into consideration through inclusion criteria stating that children will need to have sufficient cooperation and cognitive understanding to perform the tasks and access the online training program. Furthermore, we have chosen an assessment protocol that also requires enough cognitive understanding, receptive language ability and cooperation to follow instructions and complete tasks on iPad, therefore this provides an additional opportunity, in addition to discussion with families prior to recruitment, to ensure criteria are met prior to randomisation.

Need to consider and discuss how physical impairment may impact on ability to accurately assess intellectual ability, particularly for those with more significant physical impairment.

It is acknowledged that assessing intelligence in children with CP requires consideration of motor/communication and/or visual impairments. This has been taken into consideration in the following ways:

Children with GMFCS level V (most significant level of physical impairment) have been excluded from this trial.

Assessment of children will be conducted via Q-interactive – the Pearson online platform whereby assessments are administered via iPad. This format was chosen as it aligns with the inclusion criteria that participants have sufficient cooperation and cognitive understanding to access the online program and perform the tasks.

Page 6, lines 30-36: “Furthermore, by combining technological innovation with proposed models of cognitive plasticity, the possibility of accessible interventions, delivered via computers, iPads or similar devices, has emerged.” Provide references /examples.

To support this statement, I have included a reference to a recent review specifically related to the CP and DCD population and also to a published protocol of a study investigating computerised working memory training in children with CP. (page 6)

For example, Løhaugen et al.³² have proposed a randomised controlled trial assessing the efficacy of computer-based working memory training in children with cerebral palsy.

Page 7-8 – I wonder if you could provide a little more detail regarding the SMART program – is it individualised to each child and their current intellectual level or does each child undergo exactly the same program? Is it just a time-based intervention or does the child need to show competence in a given area before being able to progress?

Additional information is provided in the following paragraph introduced into page 8:

The SMART program itself consists of 55 modules that can be worked through at the participant’s own pace. Progress to each module requires successful completion of the preceding one. A maximum of five modules can be completed per day. Each module presents a proposition in the form of a relations between nonsense words, and then asks a yes/no question based on the proposition. For example, “SAJ is the same as MIS. Is MIS the same as SAJ?”. Derived relations are also trained through the addition of more than two nonsense words. For example, “SAJ is the same as MIS. QUW is the same as SAJ.” Is QUW the same as MIS?” Each module provides multiple examples of the relationship being trained, and if 16 questions are answered correctly, each within a 30-second time frame, the next module is unlocked. Additional relations trained include opposite, more than and less than.

How is the SMART program accessed? Is it something that needs to be purchased by the department administering the intervention? Can it be freely accessed? Is it available in multiple languages?

At present, the program is available in English or Dutch, and the following phrase has been added to page 7:

“currently available in English or Dutch,”

The following sentences have been introduced into the study design section to provide further information about how the program is accessed (page 10).

“Participants will be provided with log-in details enabling them to access to the online program at no cost via the program website (<http://raiseyouriq.com/>) for up to five months. They will be able to access it at home via either iPad or computer.”

Page 8 – assessment timepoints don’t need to be specified under aims, include this in method.

Thank you for your feedback. The changes have been made and this information is now in the study design subsection of the methods section.

Page 8-9 : primary hypothesis may be clearer if worded: “participants in the intervention group will demonstrate improved intellectual ability on the WISC immediately post intervention compared to a waitlist control group receiving usual care”.

Thank you for the suggestion for clarity, I have reworded the sentence on page 10:

1. Participants in the intervention group will demonstrate improved on a standardised test of intellectual ability performance immediately post-intervention when compared to a waitlist control group receiving care as usual. (Wechsler Intelligence Scale for Children - Fifth Edition; WISC-V)

Page 9-10: will demographic information/additional baseline information be collected eg. age, MACS, GMFCS, type of schooling they are attending, socioeconomic info etc. This information should be specified.

Further details have been added under Measures to provide this information (page 13):

Demographic information will be obtained via a parent survey, gathering information on the participant’s background, including gestational age, comorbid diagnoses, and GMFCS classification. Further demographic information includes school year, type of school, and whether any additional teaching support is accessed, along with parent education and household income.

If the waitlist control group will begin receiving the treatment following 20 weeks of usual care, how will long term (40 week post Rx) outcomes be compared? This may be very important comparison information in regard to long term benefits of the intervention.

We acknowledge that due to the wait-list control nature of the study, the long-term outcomes (20 weeks post completion of intervention) cannot be assessed for the wait-list control group, and that this is a limitation of the study. However, there are two reasons for this decision. Firstly, to assess

outcomes at a fourth time-point (20-weeks post intervention for the wait-list control group) will extend the duration of the study beyond the parameters of a PhD.

Secondly, a key consideration in the development of this study is the need to account for practice effects when standardised cognitive assessments are administered multiple times. To measure long-term outcomes in the wait-list control group would mean the participants in this group would be assessed on one additional occasion than the intervention group, and further practice effects would be confounding factors in the experiment.

We decided upon a wait-list control study to ensure that all families who participate in the study receive access to the intervention at some point.

Page 10 – how will “sufficient cooperation and cognitive understanding” be determined? Through standardised assessment or ability to complete baseline assessment? This should be specified so that the study could be replicated.

It is agreed that further information regarding these criteria are important for study replication. If children are able to undertake iPad-based assessment at baseline, they are deemed to meet these criteria. The following sentence has been inserted on page 11:

“Sufficient cooperation and cognitive understanding will be confirmed at baseline assessment, as participants who are able to undertake the iPad-based assessments will be deemed to meet criteria.”

Page 10, line 33. Make reference to assessment timepoints in Figure 1.

Thank you, this information has been incorporated into Figure 1.

Will information on how quickly the child completes that program be collected? I wonder if this might correlate with benefits from the intervention (motivation may correlate with outcomes) and could be interesting information given the current research regarding motivation and outcomes of intervention.

Thank you for this feedback, this information will be collected (as it is automatically recorded in the administrator version of SMART), and a section on this has been added to measures on page 17:

“The SMART program provides an internal measure of Relational Ability, and all children will complete this measure at baseline, 20 weeks and 40 weeks. In addition, information will be recorded on time taken for each participant to complete the program.”

Ensure all information on the CONSORT checklist are specified – randomisation procedure, allocation concealment, specify blinding of assessors.

Thank you for this feedback, details have now been added on page 12

“Randomisation

Baseline assessments and demographic questionnaires will be completed prior to randomisation. Once complete, participants will be randomised to either waitlist control or intervention group. Randomisation will be via stratified random blocks, using a computer-generated block randomisation sequence. Allocation to either waitlist control or intervention will be recorded on pieces of paper, and

these will be folded, then placed inside opaque envelopes by a staff member not involved in the study. Envelopes will be sealed, and only opened upon completion of baseline assessment. Participants will be stratified according to IQ (<70 or ≥70), as measured on baseline assessment.

Blinding

Given the nature of the intervention, participants will not be blinded as to which group they are assigned to. As assessment will be undertaken by the first author as part of a PhD project, assessors will not be blinded in this project.”

Outcome measurement – discuss validity and reliability of these measures in the CP population aged 8-12 years (your study population). If they are not validated in this population, that should be made clear and a note made that no valid alternatives are available. Particularly for primary outcome measure.

Thank you for bringing this important consideration to our attention. We have attempted to highlight how we approached this issue through the following inclusion on page 12

“It is noted that many of these assessments have not been validated for children with cerebral palsy, and have been chosen as no valid alternatives are available. However a review of assessments for children with cerebral palsy found that motor involvement, communication and visual impairment were key factors in determining suitability of assessments. (8) We have specifically chosen an iPad-based assessment delivery format for our primary outcome (full-scale IQ), that is similar in motor and language demand to the intervention program itself. If children are able to meet the inclusion criteria for the study, it is anticipated that they will also be able to complete the assessments.”

Page 14, who will conduct the qualitative interviews ie. Will it be the researcher who administered the intervention or completed assessments, or will it be an independent person? Do you know what qualitative design will be used? If so, it would be beneficial to detail this.

We have adjusted this section to include the relevant information. The same researcher who conducted the assessments will conduct the interviews. Page 17:

Qualitative Interview

At the conclusion of the study, semi-structured interviews will be conducted with children and caregivers by the first author, a registered psychologist. The aim of the interview is to explore participants’ engagement with the online cognitive rehabilitation program and gain qualitative insights into families’ experience with the program. Questions will cover what families like and disliked about the program, how easy they found it was to access at home, and to remain engaged. If the program is found to be effective, such qualitative insights will be valuable in the translation phase. The script for the interview can be found in Supplementary Appendix A. Interviews will be recorded, transcribed and analysed.

Page 15 – specify which statistical tests are planned

We have extended our section on statistical analysis (page 18-19)

Statistical Analysis: Study hypotheses will be analysed by means of appropriate statistical tests, with statistical significance for all tests set at $p < 0.05$ with adjustment for multiple comparisons, and all analyses will be intention to treat. We propose to carry out a Benjamini-Hochberg procedure to control for false discoveries.⁵³

Mixed analysis of covariance analyses will be conducted with time (baseline and 20 weeks) as the within subjects variable, and group (intervention or waitlist) as the between subjects variable, and baseline data as the covariate. Secondary analysis will profile cognitive change over time for participants based on their test scores. This will include t tests and linear regression to explore within-subject differences from post-intervention to follow-up, and over three timepoints (baseline, post-intervention and follow-up) for participants in the intervention group.

CONSORT – Add in when randomisation occurs. I wonder if the flowchart could be slightly modified so that timepoints and the overall timeframe of the study are slightly clearer – am I correct in assuming that there is 5 assessment timepoints given the waitlist control? T1 is the baseline (both groups), T2 is the post intervention assessment (both groups will be assessed at this timepoint?). Specify that this will be the primary endpoint of the figure. It is then unclear exactly when assessments of each group occur – I think there is probably T3 (comparison post Rx Ax), then T4 (intervention 40 weeks follow up) and T5 (control 40 week follow up). It would be great for these timepoints to be clearly reflected in the CONSORT Flowchart.

Thank you for the suggestions to improve the clarity of the flowchart. We have added in randomisation, post baseline assessment. We do not intend to assess at follow-up for the control group due to time constraints, burden on the families of a fourth assessment visit, and also due to the fact that this would be the fourth administration of cognitive assessment to the wait-list control group, an extra assessment the intervention group would not be administered. Follow-up for the waitlist control group would not be comparable to the follow-up for the intervention group due to the extra administration of tests.

Reviewer: 2

Reviewer Name: Dr. Matt C. Howard

Institution and Country: University of South Alabama, USA

Please state any competing interests or state 'None declared': None Declared

Please leave your comments for the authors below

I was asked to review the submission, "A randomized controlled trial of a novel online cognitive rehabilitation program for children with cerebral palsy: A study protocol." The submission is a proposed trial to test the effectiveness of a computer program in developing the cognitive abilities of children with cerebral palsy (CP). While I am generally familiar with the content area, both regarding the assessment of interventions as well as the rehabilitation of special populations, I must acknowledge that I do not have significant experience assessing trials. Nevertheless, I reviewed relevant material provided by the journal, and I feel that I can appropriately assess this submission.

Overall, I felt that this submission achieved the typical objectives of a proposed trial. It was clear, concise, and provided an adequate overview of the proposed study. I do have three suggestions that I believe should be implemented before the submission is accepted, however.

1.) The proposal discusses a wait-list control design. On Page 3, the submission indicates that, "No active control group is included in this study; therefore we cannot determine impact of the intervention independent of focused use of a computer program." Can't the authors determine this using the wait-list control group? I am confused regarding the intent of this statement.

Thank you for highlighting the need to clarify this limitation. We consider this a potential limitation due to concerns in the online cognitive testing literature around placebo and expectancy effects. To be clear: the wait-list control group is not an active control. It could be the case that regular additional computer use, not SMART in particular, is responsible for any intervention effect. We have included a little more information in the sentence on page 3:

"potential placebo or expectancy effects arising from focused use of a computer program."

2.) I believe that a section is needed which describes the SMART program in more detail. Is it a text-based program? Is it a picture-based program? Is it virtual reality? Of course, computer programs greatly vary, and findings regarding one program may have little to no relation with a different computer program. As the submission currently reads, I have no idea which literatures should be incorporated with the proposal trail. It would be most ideal if the authors could upload a video of the program in use as supplemental materials.

Thank you for this feedback, we have added in additional information about the program, including a link to the program's own website. As we are not the developers of the program, we feel that the website itself would be best way to access images and video of the program itself. The following paragraph has been added to page 8:

"The SMART program itself consists of 55 modules that can be worked through at the participant's own pace. Progress to each module requires successful completion of the preceding one. A maximum of five modules can be completed per day. Each module presents a proposition in the form of a relations between nonsense words, and then asks a yes/no question based on the proposition. For example, "SAJ is the same as MIS. Is MIS the same as SAJ?". Derived relations are also trained through the addition of more than two nonsense words. For example, "SAJ is the same as MIS. QUW is the same as SAJ." Is QUW the same as MIS?" Each module provides multiple examples of the relationship being trained, and if 16 questions are answered correctly, each within a 30-second time frame, the next module is unlocked. Additional relations trained include opposite, more than and less than.}

3.) The authors use a wait-list control group design, but they cite prior research that convincingly argues that the intervention is effective. Also, as it is a computer program, I would assume that the program can be extended to all participants fairly easily. In other words, the use of a wait-list control group does not seem to be a logistical necessity, and it is instead solely for the test of the intervention. Can the authors provide any assurance that the delayed administration to wait-list participants will not negatively affect them? Otherwise, I would see the use of a wait-list control design as causing undue harm to the participants.

Pilot studies of the online cognitive intervention program have provided encouraging data that suggests this program could be effective, with increases in performance in cognitive assessments of a standard deviation or more. However, we feel that it is very important to be able to account for

practice effects in the repeated administration of cognitive tests such as the WISC-V when evaluating the effectiveness of cognitive interventions, in order to determine such programs' efficacy, prior to recommending them as interventions that may involve costs to families of significant amounts of time, as well as potentially financial costs, that may otherwise be directed to alternative interventions. We have added further information on page 9, discussing the need to account for practice effects.

While promising, practice effects need to be accounted for when repeated administration of standardised measures of intelligence occurs, as they may influence performance, with average gains of 6-7 points over a one-month period found for the WISC-V measure of full-scale IQ 39. Furthermore, assessment of fluid reasoning ability may be more affected by practice effects than verbal or working memory tasks 39, as fluid reasoning tasks are associated with ability to solve novel problems. A randomised controlled design rather than pre- and post-intervention studies could control for practice effects, but the two studies that have utilised this design have been limited by small sample sizes and high attrition rates. 40, 41 To date, no studies have investigated the efficacy of SMART in the cerebral palsy population.

Reviewer: 3

Reviewer Name: Chris Jones

Institution and Country: Brighton and Sussex Medical School, UK

Please state any competing interests or state 'None declared': None declared.

Please leave your comments for the authors below

The objectives of the study are clearly described, but the statistical analysis section needs to be rewritten to make it clear that the sample size and planned analysis fit with the objectives. Information about randomisation and data management are also required.

The primary analysis is a comparison of the intervention and control groups "immediately post-intervention" - i.e. at 20 weeks. No comparison can be made at 40 weeks, because at this point both groups will have received the intervention. I'm not sure what the point of the assessments at week 40 is, as comparing the groups at this point doesn't make sense for the objectives of the study.

We acknowledge that due to the wait-list nature of the study design (chosen in the interests of equity to allow all families access to the intervention within a reasonable time frame), the key analysis to compare the intervention with the control group is the immediate post-intervention 20-week comparison. The primary analysis will be to examine the efficacy of the intervention from baseline to post-treatment with a mixed between- and within-groups analysis. 20-week follow-up will be a secondary analysis for the intervention group alone. We had decided against a 20-week follow-up assessment for the wait-list control group partly due to the burden of further assessment on the families, and also due to the additional practice effects this group would be subject to, completing assessment a fourth time. We acknowledge this as a limitation of the study.

Despite this, the analysis method suggested is a repeated measures ANOVA using data from baseline, week 20 and week 40. Since the situation at week 40 is different, and week 40 is not mentioned as part of the primary objective, it should not be included in the analysis. A more appropriate approach would be to perform linear regression for WISC-V at 20 weeks, with intervention group and WISC-V at baseline as independent variables.

We have carefully considered this feedback and agreed that week 40 outcomes are not the primary objective and that we need to clarify the analyses to be used. Therefore, we agree the primary analysis will be to compare the group outcomes at 20 weeks. However, we consider that a repeated measures ANCOVA with baseline data as the covariate does address the primary objective, as the time*group interaction obtained from this analysis will allow a comparison of mean change from pre to post on outcomes between the two groups.

Secondary analyses, including t tests assessing change from post to follow-up, or linear regression to look at change over three time-points, will be performed on outcomes from the intervention group to assess 40-week outcomes. We have adjusted the paragraph to read (page 18-19):

Mixed analysis of covariance analyses will be conducted with time (baseline and 20 weeks) as the within subjects variable, and group (intervention or waitlist) as the between subjects variable, and baseline data as the covariate. Secondary analysis will profile cognitive change over time for participants based on their test scores. This will include t tests and linear regression to explore within-subject differences from post-intervention to follow-up, and over three timepoints (baseline, post-intervention and follow-up) for participants in the intervention group.

Generally this section needs to be re-written by a statistician for clarity. The sentence "Study hypotheses will be analysed by means of appropriate statistical tests, with statistical significance for all tests set at $p < 0.05$ with adjustment for multiple comparisons, and all analyses will be intention to treat." lumps several important statistical concepts together in a way that adequately describes none of them.

Thank you for your feedback, we have outlined steps taken below to address all of these concerns separately and to better describe the statistical analysis.

What multiple comparisons are being adjusted for? How is this adjustment being made, and was this taken into account for the sample size calculation?

Given the number of multiple comparisons likely, due to secondary outcomes being assessed, we believe that controlling the false discovery rate will be more suitable than a Bonferroni adjustment. Therefore, the following sentence has been added (page 18):

We propose to carry out a Benjamini-Hochberg procedure to control for false discoveries.(9)

The sample size calculation should be described in a separate section. It likely needs to be done differently as it is based on what appears to be an inappropriate analysis method. Not enough information is provided to recreate the current calculation - all assumptions made need to be clearly stated, as does the software used.

The following section has been added to more clearly provide this information (page 19)

"Power Analysis and Sample Size:

Power analysis was conducted using the software package G*Power (10), for an ANCOVA repeated measures, between factors test. With a sample size of 60 subjects, and assuming an error rate of 5% and $p=0.70$ for within-subject correlations, this sample size results in 81.37% power to detect a large (Cohen's $f=0.40$) mean difference of 12 IQ points between groups, after allowing for an attrition rate of 10%."

What is the justification for using an effect size of 0.7? Is this based on any previous data?

A large effect size is equivalent, given our planned sample size, to an increase in IQ of approximately 12 points, just under a standard deviation for IQ. ($M=100$, $SD=15$) We believe this is justified given the results from pilot studies of increases in excess of one standard deviation (11)

What effect size would be considered medically significant?

As increases of around .5 of a SD can occur due to practice effects when administering cognitive tests (12) we feel that an effect size closer to a standard deviation is more appropriate when considering clinical significance.

What attrition rate is expected?

By aiming to recruit 60 participants, we have allowed for an attrition rate of 10%. This information has now been included in the power analysis and sample size section.

The randomisation method used needs to be stated clearly. It's probably simple randomisation, but it could be stratified by GMFCS for example - it needs to be clear. Also, how was the randomisation list generated? - block sizes, software used etc. How does it work in practice?

The following section has been added to clearly indicate the randomisation method used (page 11-12)

"Randomisation

Baseline assessments and demographic questionnaires will be completed prior to randomisation. Once complete, participants will be randomised to either waitlist control or intervention group. Randomisation will be via stratified random blocks, using a computer-generated block randomisation sequence. Allocation to either waitlist control or intervention will be recorded on pieces of paper, and these will be folded, then placed inside opaque envelopes by a staff member not involved in the study. Envelopes will be sealed, and only opened upon completion of baseline assessment. Participants will be stratified according to IQ (<70 or ≥ 70), as measured on baseline assessment."

Data management - no information is given with regards to data management - what database software will be used and where will the data be stored?

The following paragraph has been added to address this feedback (page 12)

"Participants will be allocated randomly generated identification codes, and these will be used to de-identify hardcopy and electronic files. Paper copies relating to assessment will be de-identified and physically stored in a locked filing cabinet at the Queensland Cerebral Palsy and Rehabilitation Research Centre. Electronic data will be stored on REDCap, a secure web platform for creating and

managing online databases. The installation of REDCap used for this project is hosted by The University of Queensland and managed by the Queensland Clinical Trials and Biostatistics Centre.”

VERSION 2 – REVIEW

REVIEWER	Michelle Jackman John Hunter Children's Hospital Newcastle Australia
REVIEW RETURNED	06-Apr-2019

GENERAL COMMENTS	Authors have addressed most feedback, I would encourage authors to review wording of hypothesis 1 as the sentence is currently not clear: 1. Participants in the intervention group will demonstrate improved on a standardised test of intellectual ability performance immediately post-intervention when compared to a waitlist control group receiving care as usual. (Wechsler Intelligence Scale for Children - Fifth Edition; WISC-V) I wish the authors good luck with their study!
---

REVIEWER	Dr. Matt C. Howard University of South Alabama, USA
REVIEW RETURNED	06-Apr-2019

GENERAL COMMENTS	I was asked to review a resubmission of, "A randomized controlled trial of a novel online cognitive rehabilitation program for children with cerebral palsy: A study protocol". My original comments were minimal, and the authors have done a fine job to address them. I believe that the current version of the manuscript should be published. If any additional changes should be made based on my original comments, I would perhaps add a sentence or two to the "Adverse events" section to assures readers that the wait-list control group would receive the intervention if they are shown to be substantially impacted by its omission. Thank you for allowing me to read your work.
--

REVIEWER	Chris Jones Brighton and Sussex Medical School, UK
REVIEW RETURNED	29-Mar-2019

GENERAL COMMENTS	My previous comments have been addressed.
---

VERSION 2 – AUTHOR RESPONSE

Thank you for the opportunity to respond to reviewers' feedback. We have made the revisions suggested by Reviewers 1 and 2. On page 9, we have corrected Hypothesis 1 to address Reviewer 1's recommendation. It now reads:

"Participants in the intervention group will demonstrate improved performance on a standardised test of intellectual ability immediately post-intervention when compared to a waitlist control group receiving care as usual. (Wechsler Intelligence Scale for Children - Fifth Edition; WISC-V)42, 43"

On page 12, we have added the following sentence to address Reviewer 2's feedback under Adverse Events:

"All participants allocated to the wait-list control group receive access to the intervention after the second assessment. This ensures no adverse impacts through omission of intervention in the event that the intervention is found to be efficacious."